# Analysis of Systemic Inflammatory Factors and Survival Outcomes in Endometrial Cancer Patients Staged I-III FIGO and Treated with Postoperative External Radiotherapy

**DOI:** 10.3390/jcm9051441

**Published:** 2020-05-12

**Authors:** Katarzyna Holub, Fabio Busato, Sebastien Gouy, Roger Sun, Patricia Pautier, Catherine Genestie, Philippe Morice, Alexandra Leary, Eric Deutsch, Christine Haie-Meder, Albert Biete, Cyrus Chargari

**Affiliations:** 1Radiation Oncology Department, Hospital Clinic de Barcelona, University of Barcelona, 08036 Barcelona, Spain; abiete@clinic.cat; 2Radiotherapy Department, Gustave Roussy Cancer Campus, 94800 Villejuif, France; fabiobusato2004@libero.it (F.B.); roger.sun@gustaveroussy.fr (R.S.); Eric.DEUTSCH@gustaveroussy.fr (E.D.); christine.haiemeder@gustaveroussy.fr (C.H.-M.); cyrus.chargari@gustaveroussy.fr (C.C.); 3Surgery Department, Gustave Roussy Cancer Campus, 94800 Villejuif, France; sebastien.gouy@gustaveroussy.fr (S.G.); philippe.morice@gustaveroussy.fr (P.M.); 4Medical Oncology Department, Gustave Roussy Cancer Campus, 94800 Villejuif, France; patricia.pautier@gustaveroussy.fr (P.P.); Alexandra.LEARY@gustaveroussy.fr (A.L.); 5Pathology Department, Gustave Roussy Cancer Campus, 94800 Villejuif, France; catherine.genestie@gustaveroussy.fr

**Keywords:** endometrial cancer, systemic inflammation, neutrophil-to-lymphocytes ratio (NLR), systemic immune-inflammatory index (SII), monocyte-to-lymphocyte ratio (MLR), lymphopenia

## Abstract

Background: The causal link between elevated systemic inflammation biomarkers and poor survival has been demonstrated in cancer patients. However, the evidence for this correlation in endometrial cancer (EC) is too weak to influence current criteria of risk assessment. Here, we examined the role of inflammatory indicators as a tool to identify EC patients at higher risk of death in a retrospective observational study. Methods: A total of 155 patients surgically diagnosed with EC stage I-III FIGO 2009 and treated with postoperative External Beam Radiotherapy (EBRT) ± brachytherapy and chemotherapy according to ESMO-ESTRO-ESGO recommendation for patients at high risk of recurrence at the Gustave Roussy Institut, France, and Hospital Clínic, Spain, between 2008 and 2017 were evaluated. The impact of pre-treatment Neutrophil-to-Lymphocyte Ratio (NLR ≥ 2.2), Monocyte-to-Lymphocyte Ratio (MLR ≥ 0.18), Systemic Immune-Inflammatory Index (SII ≥ 1100) and lymphopenia (<1.0×10^9^/L) on overall survival (OS), cancer-specific survival and progression-free survival was evaluated. Subsequently, a cohort of 142 patients within high-advanced risk groups according to ESMO-ESGO-ESTRO classification was evaluated. Results: On univariate analysis, NLR (HR = 2.2, IC 95% 1.1–4.7), SII (HR = 2.2, IC 95% 1.1–4.6), MLR (HR = 5.0, IC 95% 1.1–20.8) and lymphopenia (HR = 3.8, IC 95% 1.6–9.0) were associated with decreased OS. On multivariate analysis, NLR, MLR, SII and lymphopenia proved to be independent unfavorable prognostic factors. Conclusions: lymphopenia and lymphocytes-related ratio are associated with poorer outcome in surgically staged I-III FIGO EC patients classified as high risk and treated with adjuvant EBRT and could be considered at cancer diagnosis. External validation in an independent cohort is required before implementation for patients’ stratification.

## 1. Introduction

Endometrial cancer (EC) is the most common malignancy of the female reproductive system in developed countries and the only gynecological cancer with a rising incidence (+1.3% per year over the last 10 years) and mortality rate [1,2]. It is estimated that in 2020, 65,620 new patients will be diagnosed, and 12,590 women will die of this disease in the United States [1]. Although the average 5-year survival is high (95.0%) when the disease is confined to the uterus, it is substantially lower in the presence of regional or distant dissemination [2,3].

The assessment of EC recurrence risk relies on the consensus of the European Society for Medical Oncology (ESMO), the European Society for Radiotherapy & Oncology (ESTRO) and the European Society of Gynecological Oncology (ESGO), which standardized EC primary treatment choice by using traditionally applied tumor-related risk factors such as tumor grade, histology, clinical and radiological stage according to the International Federation of Gynecology and Obstetrics (FIGO) and lymphovascular invasion [4]. However, there is growing evidence that the outcome of EC patients may also depend on factors beyond the classically established risk indicators mentioned above [5,6]. Up to now, the only attempts to refine the ESMO-ESGO-ESTRO risk assessment have been directed towards surgical nodal staging, and new prognostic tools beyond this classification are warranted in order to identify women with a higher risk of death [7]. The role of a host’s inflammatory response in cancer progression has been the topic of numerous research articles published in the last years, which analysed the level of inflammatory ratios based on blood components, such as Neutrophil-to-Lymphocytes Ratio (NLR), Platelet-to-Lymphocytes Ratio (PLR), Monocyte-to-Lymphocytes Ratio (MLR) or Systemic Immune-Inflammation Index (SII) as useful prognostic factors in different malignancies [7,8,9,10,11,12,13,14,15,16]. Nevertheless, in the case of EC, there is insufficient evidence to support their use in clinical practice, as shown by Ethier et al. in an extensive review of 26 studies with a total of 10,530 patients with different gynecological cancers published in 2017, where EC was depicted only in two studies with less than 1000 patients [17,18,19]. Moreover, none of these inflammatory biomarkers have so far been proposed to refine the ESMO-ESGO-ESTRO EC risk assessment [17,18,19,20,21,22,23,24].

Our research aims to provide new data about the suitability of inflammatory biomarkers as indicators of EC prognosis.

## 2. Patients and Methods

### 2.1. Patients’ Inclusion Criteria

The institutional databases of patients with histologically proven EC, surgically staged as FIGO I-III and treated between February 2008 and January 2017 with postoperative External Beam Radiotherapy (EBRT), were retrospectively reviewed. The study protocol was approved by our Institutional Review Boards (the Gustave Roussy Multidisciplinary Board for Gynecological Cancer and the Ethical Committee of Hospital Clinic de Barcelona with project identification code: Hb_endometri) and was conducted in accordance with ethical standards following the rules of the Declaration of Helsinki (https://www.wma.net/what-we-do/medical-ethics/declaration-of-helsinki/), revised in 2013.

The clinical data of women with high-intermediate risk, high risk, or advanced uterine endometrial cancer (according to ESMO-ESGO-ESTRO risk stratification) and with accessible results of blood test done within 3 months prior to surgery were selected and retrospectively examined in two European institutions. All patients underwent hysterectomy with bilateral adnexectomy and pelvic and/or para-aortic lymphadenectomy according to individual staging requirements. The indication of postoperative adjuvant treatment was applied according to the ESMO guidelines [4]. Patients diagnosed with uterine sarcomas, acute or chronic infections (including Human Immunodeficiency Virus and virus of hepatitis), immunodeficiency, other active malignancies, hematological disorders, patients treated with therapies causing immunomodulation (such as steroid or anti-inflammatory drugs), patients not treated with adjuvant EBRT or patients with metastatic disease were excluded. A flowchart for patients’ exclusion is shown in Appendix A.

### 2.2. Inflammatory Indicators

The following preoperative inflammatory indicators were examined: Neutrophil-to-Lymphocyte ratio (NLR: neutrophil count divided by the lymphocyte count), Systemic Immune inflammation Index (SII: neutrophil count multiplied by platelets count and divided by the lymphocyte count) and Monocyte-to-Lymphocyte ratio (MLR: monocyte count divided by the lymphocyte count).

From the entire cohort, the most appropriate cut-off value for each indicator was chosen, according to the optimal decision threshold from the receiver operating characteristics (ROC) curve. The main endpoint of our research was to examine overall survival (OS) and cancer-specific survival (CSS) of the entire cohort stratified according to the level of inflammatory factors. The impact of inflammatory factors on Progression Free Survival (PFS) was also assessed. In addition, biological ratios were examined for correlations with age, FIGO stage, tumor grade and histology.

### 2.3. Follow-Up and Statistical Analysis

Follow-up (FU) was performed as a clinical exam every 3–4 months for the first 2 years, and then every 6 months for the next 3 years (with radiological exams guided by symptoms), following ESMO guidelines [4]. All statistical tests were two-sided and *p*-value < 0.05 was considered as statistically significant. All examined factors were defined as binary variables, by finding the cut-off value from a ROC curve, and their balance across prognostic characteristics was assessed using Chi-square test (X^2^). Frequencies were compared using Fisher’s exact test for categorical variables. Statistical significance was calculated by Student’s *t* test when comparing two groups, or by one-way or two-way ANOVA when comparing three or more. Survival outcomes were calculated from the date of surgery (first treatment) to the event occurrence, which was the death by any cause for OS and the cancer-related death for CSS. Progression-Free-Survival was calculated from the date of surgery to the disease progression or cancer-related death. Patients were censored if no event occurred. Survival curves for different types of survival measures were constructed via the Kaplan-Meier method and comparisons were made using the Wald test. For univariate and multivariate analyses of prognostic variables, Cox proportional hazards regression was applied and the variables with *p*-value < 0.10 in Cox univariate analysis were analysed in multivariate analysis. All statistical analyses were performed using standard software (SPSS v 23.00; SPSS Inc., Chicago, IL, USA).

## 3. Results

### 3.1. Patients and Tumors

A total of 155 women (77 patients in Gustave Roussy Cancer Center and 78 patients in Hospital Clinic de Barcelona) were eligible for inclusion in the study. The median age at diagnosis was 63.1 years (range 27.9–98.9 years). The most frequent histology was endometrioid carcinoma (61.3%). After surgery, the disease stage was classified according to the FIGO 2009 as stage I-II in 97 patients (62.6%) and stage III in 58 cases (37.4%). High tumor grade (grade 3) was present in 96 patients (61.9%). Lymphadenectomy was performed in 129 patients (83.2%), pelvic in 30 patients (19.4%), paraaortic in 21 patients (13.5%), and both in 78 patients (50.3%). Out of the remaining 26 patients (16.8%), 13 were classified within a high-intermediate risk group 3, 9 showed a negative result of sentinel lymph node procedure and in 4 cases these data were missing. According to ESMO-ESGO-ESTRO risk groups, 13 patients (8.4%) had high-intermediate risk cancer, 134 (86.5%) had high-risk disease, and eight (5.2%) had advanced stage. Patients’ demographics and characteristics of tumours for each center and for the entire cohort are detailed in Appendix A.

### 3.2. Treatments

All patients were treated with postoperative EBRT (mean dose was 45.4 Gy). A High-Dose Rate Brachytherapy boost (HDR-BT) was administered to 145 patients (93.5%) with a mean dose of 10 Gy for HDR-BT, 10 patients (6.2%) received exclusive EBRT (mean dose 45 Gy). A total of 78 patients (50.3%) also received adjuvant chemotherapy (CT), mostly cisplatin-based (40 patients) for 4–6 cycles, carboplatin-taxol (34 patients), carboplatin (1 patient), cisplatin-paclitaxel (1 patient), cisplatin-doxorubicine (2 patients).

### 3.3. Outcome

Median FU of the entire cohort was 46.5 months (range 4.0–108.0). Forty patients (25.8%) had died at the time of data collection, 35 of these deaths were cancer-related (22.6%). In the entire study population, 3- and 5-years OS rates were 81.0% and 73.0% respectively. Forty-seven patients (30.3%) presented disease recurrence, with median time from surgery to cancer progression of 13.0 months (range 0.7–72.5). In details, sites of recurrence were as follows: four patients presented only local (vaginal) progression, five patients had exclusively pelvic nodal progression, six patients had both local and pelvic progression, four patients had local and distant failure, eight patients had pelvic and distant progression and 20 patients had exclusively distant progression.

### 3.4. Systemic Inflammation Markers: Cut-Off Value

The most appropriate cut-off value for each indicator was as follows: 2.2 for NLR, with an area under curve (AUC) of 0.641, a sensitivity (Se) of 0.8 and a specificity (Sp) of 0.4. The optimal cut-off value for SII was 1100.0 (AUC 0.540, Se 0.4, Sp 0.7) and 0.2 for MLR (AUC 0.541, Se 0.7, Sp 0.5). A high pre-treatment NLR ≥ 2.2 was observed in 93 patients (60.0%), SII ≥ 1100 in 30 patients (19.4%), MLR ≥ 0.18 in 125 patients (80.6%), and lymphopenia in 11 patients (7.1%). Additionally, a high level of monocytes (≥ 0.8 x10^9^/L) and platelets (≥410 × 10^9^/L) were detected in 7 (4.5%) and 8 patients (5.2%), respectively. Strong correlations were observed between all immune parameters (*p* < 0.001) in the Spearman’s test. Neutrophils count was not significant (*p* = 0.972 for a cut-off value at 7 G/L).

### 3.5. Survival Analysis

Prognostic factors in Kaplan-Meier survival analysis; univariate cox regression is shown in Table 1. Univariate analyses showed an increased risk of death in patients with high NLR (HR = 2.2, IC 95% 1.1–4.7), high SII (HR = 2.2, IC 95% 1.1–4.6), high MLR (HR = 5.0, IC 95% 1.1–20.8) and lymphopenia (HR = 3.8, IC 95% 1.6–9.0). Of note, age, FIGO stage III, tumor grade 3 and non-endometrioid histology did not reach statistical significance. CT use had no impact on mOS (*p* = 0.614) in Kaplan-Meier analysis, with 24.4% deaths in CT cohort vs. 27.3% in non-CT cohort. Regarding the possible influence of age on inflammatory factors, we did not observe any significant impact of age < 50 and age < 55 on mOS (*p* = 0.350 and *p* = 0.812), nor any significant correlation with levels of NLR, SII, MLR or lymphopenia. Menopausal status was not available, and could therefore not be tested accurately.

Given the correlation between parameters, separate multivariate analyzes were done for each factor. On multivariate analysis for OS, the following variables were independent factors for OS: NLR, MLR, SII and lymphocytes count (Table 1). The effect of inflammatory markers on OS is shown in Figure 1. The same analysis for CSS revealed similar results for FIGO stage III, MLR and lymphopenia (Table 2). The effect of inflammatory markers on CSS is shown in Figure 2. Regarding PFS, only FIGO stage III and pre-treatment lymphopenia were statistically significant (Table 3).

## 4. Discussion

The impact of systemic inflammation on cancer development has attracted a great deal of attention over the last years [7,8,9,10,11,14]. The analysis of the hematological inflammatory indicators and their ability to serve as possible prognostic factors has been investigated in several malignant tumours [12,13,15,16,17,18,19,20,21,22,23,24]. These new prognostic indicators are warranted to refine the prognosis and guide adjuvant treatments because the survival outcomes of patients treated according to the guidelines based only on tumor characteristics may differ widely among patients classified within the same group of risk. We observed through the survival analysis, that a high pre-treatment NLR, SII, MLR and a low lymphocytes count were associated with a worse OS for EC patients surgically staged as FIGO I-III and who received adjuvant EBRT. Due to the correlation existing between all inflammatory factors, the multivariate analysis model was evaluated separately for each immune factor studied. NLR, SII, MLR, and lymphocytes count were significant for OS.

These data are in line with other recently published results in the literature. Dong et al. reviewed clinical records of 510 Chinese EC patients surgically treated between 2010-2016 and found that NLR was an independent prognostic marker for OS (HR 4.7; 95% CI, 1.5–14.1; *p* = 0.006), CSS (HR 3.6; 95% CI, 1.1-11.5; *p* = 0.028) and disease free survival (DFS, HR 2.3; 95% CI, 1.0-5.2; *p* = 0.044) [25]. Based on the data of 101 patients, Mirili et al. confirmed that NLR > 3.3 and PLR>177 were associated with shorter PFS and OS, and were the first to prove that SII > 1035.9 and prognostic nutritional index (PNI) < 38 were also independent prognostic factors for worse survival outcomes in EC. The authors also examined a correlation between inflammatory factors and classically used prognosticators such as lymph node involvement, FIGO stage, lymphovascular invasion, and cervical stromal invasion, which they found that were associated with higher NLR, SII, and lower PNI. Moreover, NLR and PNI were associated with ECOG performance scores 2–3 and myometrial invasion [26]. Similarly, a recent Japanese study on 197 EC patients identified a high NLR and PLR as predictive of lymph node involvement, observed in 25 patients (13%) [27]. Acikgoz et al. reported that the preoperative NLR > 2.41 was a significant predictor for cervical stromal involvement in endometrioid endometrial adenocarcinoma (*p* = 0.006, OR = 2.03) [20]. In an article based on the retrospective data of 320 patients, Hannuma et al. confirmed that pre-treatment NLR was an independent predictor of poor prognosis in EC (HR 3.3; 95% CI 1.2–9.5; *p* = 0.026) [18]. In a systematic review of eleven studies, elevated NLR was related with advanced stage of disease in EC patients [28]. Regarding the impact of inflammatory indices on time from surgery to disease progression, only lymphopenia at cancer diagnosis and FIGO stage III were significantly associated with poorer PFS. This may be related with the fact that 11 patients who had progression during follow-up were classified as disease-free after salvage treatment of recurrence with RT ± BT. These findings may also explain that among studies investigating the impact of inflammatory markers on EC survival outcomes, all focus on OS and very few investigated impacts on PFS [17,18,19].

The strength of our study in comparison with previously reported publications relies on its bi-institutional cohort of high-risk patients, all of them treated with EBRT. Until now, the only multicenter study with a larger cohort than ours, with 605 patients, was published by Cummings et al., but only 33% were treated with radiotherapy. Cummings et al. also included patients in FIGO stage IV [17], while all our patients were staged as FIGO I–III but classified within the ESMO-ESGO ESTRO 3–5 groups. For this reason, patients treated with exclusive HRD-BT did not meet inclusion criteria, as in accordance with the guidelines published by ESMO-ESGO-ESTRO in 2015, only patients belonging to the high-intermediate, high and advanced risk group should be treated with postoperative adjuvant EBRT ± BT (patients with low and intermediate risk of recurrence were excluded from the study) [4].

Regarding the cut-off values, the applied thresholds vary widely among studies, as there are no clinically obvious cut-points. For example, Cummings demonstrated that NLR ≥ 2.4, but not MLR ≥ 0.19 had independent prognostic significance for OS and CCS [17]. We applied similar cut-off values (NLR ≥ 2.2, MLR ≥ 0.18) based on the ROC curves for OS in our population. Although this method is not free of potential bias (overestimation of the effect), the ROC curve analysis is nowadays applied in the majority of studies on inflammatory ratios [19,26,29]. Moreover, for NLR, which is the most frequently studied ratio, our thresholds are similar to the cut-points found in the general population and match the upper limits of physiological values published by Fest et al. based on data of more than 8700 healthy subjects [30].

The classically applied variables in our study such as histology (endometrioid vs. others) and tumor grade (grade 3 vs. grade 1–2) did not achieve statistical significance in univariate Cox regression. We assume that adjuvant treatments may have alleviated the effects of such clinical features by decreasing their intrinsic prognostic value. Moreover, all our patients belong to the high-risk cohort (ESMO-ESGO-ESTRO 3–5 group), where these classical parameters should not be analysed separately but as a sum of characteristics. For example, the tumor grade 3 is a stronger prognosticator of more aggressive evolution than FIGO stage or histology [4].

The lymphocyte count was the most significant parameter for survival. Given the fact that lymphocyte count is part of all indexes, we may hypothesize that lymphopenia could be the most robust biomarker among all tested factors, that all are depending on lymphocytes count. In line with this, only the significance of lymphocytes count remained if indexes cutoff were not chosen from ROC curves, but instead based on a value used in daily routine. Interestingly, we however observed that the percentage of patients showing a high level of blood test components (established according to values in our laboratory: neutrophilia ≥ 7.0, monocytosis ≥ 0.8, thrombocytosis ≥ 410 or lymphopenia < 1.0) was relatively low (11.6%, 4.5%, 5.2% and 7.1% of patients, respectively) and does not explain the high number of patients with elevated corresponding inflammation markers (high NLR in 60.0%, high MLR in 80.6% and high SII in 19.4% respectively). In particular, the prevalence of neutrophilia at diagnosis was much lower, as compared to previous cohorts of patients with locally advanced squamous cell carcinoma (e.g., cervix, head and neck, anal carcinoma), where neutrophilia was reported in around 25% of cases and was found to be a major prognostic factor [31,32,33,34,35,36,37]. In this study, neutrophilia was not significant for survival. In a larger series of 508 patients, Takahashi et al. showed the prognostic significance of systemic neutrophil in surgically treated endometrial cancer patients. Indeed, neutrophilia was detected in 8.3% of patients at the time of the initial diagnosis and was found to be significantly associated with an advanced clinical stage, LVSI, and shorter survival. In details, neutrophilia was present in 5.3%, 11.1%, 10.6%, and 41.2% of the patients with stage I, stage II, stage III and IV disease, respectively [38]. These results are in line with our data, as 62.6% of our patients were diagnosed in FIGO stage I–II disease. The absence of significance in our series may be consecutive to a selection of a rather homogeneous cohort without stage IV patients. This lower incidence of neutrophilia at diagnosis, as compared with squamous cell carcinoma, may be explained by different biologies. In addition to the fact that EC tumors are usually less necrotic tumors than cervical cancer (therefore leading to less systemic inflammation), we may hypothesize that the immunomodulating effect of inflammatory indices in EC patients may be seen not only as a consequence of a high or low level of their components, but possibly as depending on the interaction between the levels of different circulating blood cells. This interaction may influence the tumor microenvironment towards the conditions favorable for cancer development or suppression. This information may provide some rationale for use of the immunotherapy treatments, given the possible influence of the tumor microenvironment on the immune checkpoint inhibitors [39,40].

We assume limitations of our work, such as its retrospective nature, although almost all studies dealing with the systemic inflammation indicators are also retrospective [16,17,18,19,20,21,22,23,24,25,26,27,28,29]. As a bi-center project, some slight differences regarding the radiotherapy schedules between the two centers were admitted, but they did not seem to influence the survival outcomes. Most of the patients included (91.6%) belonged to the high-risk group, as a consequence of inclusion criteria. Unfortunately, we did not have the records of patients’ weight and height for the majority of Spanish patients, and therefore we were unable to assess the relationship between the Body Mass Index (BMI) and inflammatory factors. We also had some difficulties in accessing the results of the blood test at cancer diagnosis, as the majority of the patients were operated in their local centers then referred for adjuvant treatment in our comprehensive cancer centers. Furthermore, it would have been methodologically stronger to assess the effect of ratios in one cohort, then validate in another. Such analyses were done, but significance was not reached, possibly because of a low number of patients.

Taking into consideration these limitations, this study is the first describing such a high number of inflammatory indicators in the same rather homogenously treated multicenter cohort of 155 EC patients. We observed that a high level of NLR, MLR and lymphopenia were associated with poorer outcome in EC and these findings are in accordance with previous similar studies [20,21,22,23,24,25,26,27]. There is still much uncertainty about how to select patients with EC who may benefit from treatment intensification. The most convincing evidence from the PORTEC3 trial concerns stage III patients and the data generated here suggest that immune parameters may also be relevant for decision-making process. Nevertheless, the immune indicators should be verified in further prospective larger studies in order to confirm the cut-off points of immune indicators before moving into widespread clinical use. Furthermore, the validation of inflammatory factors in correlation with molecular diagnosis of EC should be done. Given the demonstrated value of molecular patterns in endometrial cancer (p53 mutation, POLE I mutational analysis, mismatch repair deficiency), next step will be to assess how inflammatory indicators may be used to refine patients prognosis and/or guide adjuvant therapies based on a molecular risk classification [41]. Especially for endometrial cancer with microsatellite instability (found in approximately 30% of all cases), which are histologically characterized by presence of tumor-infiltrating lymphocytes and high immunogenicity and are particularly appropriate for immunotherapy [42]. Unfortunately, MSI status was not available for analysis in our study (which is another limitation) but should be examined further with focus on inflammatory ratio and lymphocytes count for potential correlations.

## 5. Conclusions

Pre-treatment inflammatory indicators (NLR ≥ 2.2, SII ≥ 1100, MLR ≥ 0.18 and lymphopenia) were found associated with worse survival outcome in high-risk EC patients surgically staged as I-III FIGO and treated with adjuvant EBRT. Among those, the lymphocyte count appeared as the most significant parameter. All these parameters are of interest but further research in larger cohort is required. Our study outlines the clinical importance of host-derived factors regulating the systemic inflammatory response in high-risk EC patients, and suggests implementation of this information for developing new therapeutic approaches and follow-up recommendations. Prospective external validation in an independent cohort is however required before implementing these indicators in patients risk stratification.

## Figures and Tables

**Figure 1 jcm-09-01441-f001:**
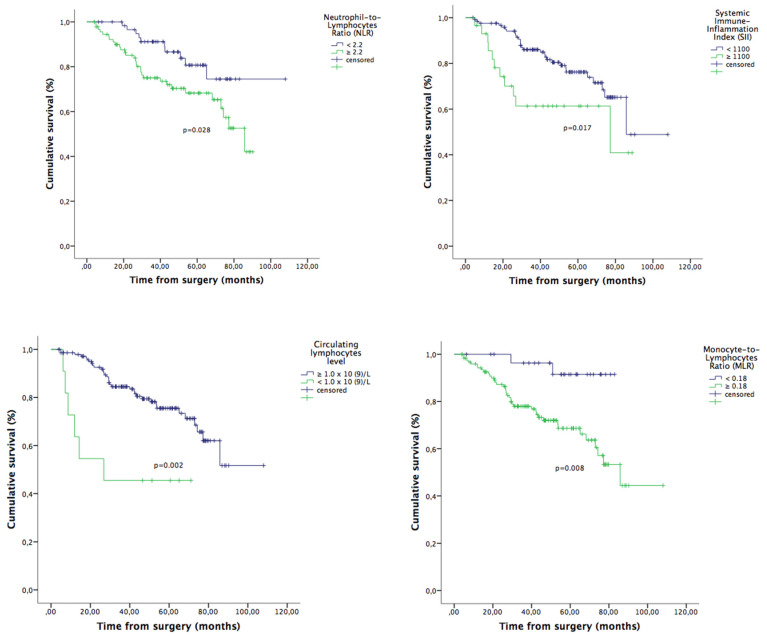
Overall survival in all patients included (*n* = 155): Impact of pre-treatment Neutrophil-to-Lymphocyte (NLR; cut-off ≥ 2.2), Systemic Immune-Inflammatory Index (SII; ≥1100), Monocyte-to-Lymphocyte Ratio (MLR; ≥0.18), and circulating lymphocytes (<1.0 × 10^9^/L) on overall survival (OS) of the endometrial cancer (EC) patients.

**Figure 2 jcm-09-01441-f002:**
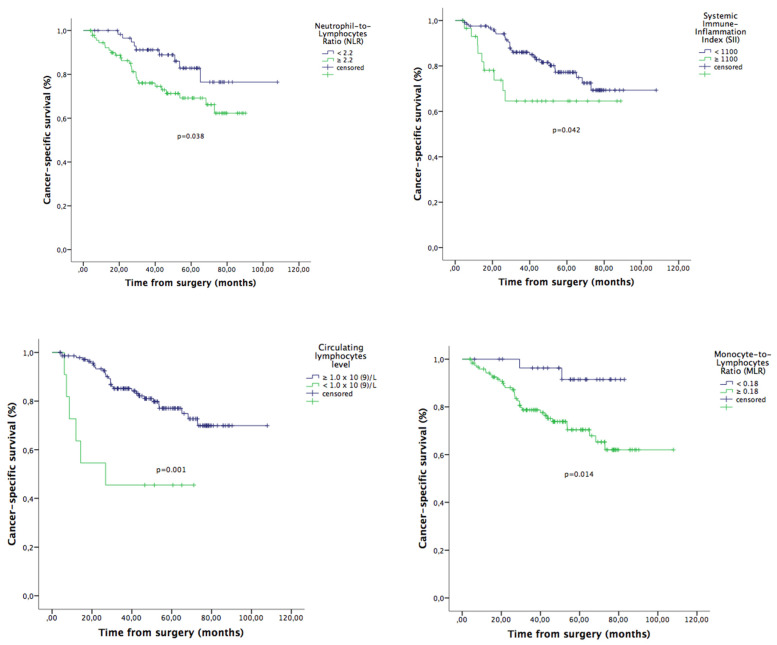
Cancer-specific survival in all patients included (*n* = 155): Impact of pre-treatment Neutrophil-to-Lymphocyte (NLR; cut-off ≥ 2.2), Systemic Immune-Inflammatory Index (SII ≥ 1100), Monocyte-to-Lymphocyte Ratio (MLR; ≥0.18), and circulating lymphocytes (<1.0 × 10^9^/L) on cancer-specific survival (CSS) of the of the endometrial cancer (EC) patients.

**Table 1 jcm-09-01441-t001:** Overall survival analysis (*n* = 155).

OS, *n* = 155 Variables (*n*)	Kaplan—Meier Survival Analysis			UNIVARIATE Cox Regression		MULTIVARIATE Cox Regression	
	X^2^	*p*-Value	OS (Range)	HR (IC 95%)	*p*-Value	HR (IC 95%)	*p*-Value
Age ≥ 65years (vs. <65 years)	0.9	0.343	67.5 (58.9–76.1) vs. 78.0 (65.9–90.1)	1.6 (0.8–3.1)	0.171	-	-
FIGO stage III (vs. stage I–II)	2.1	0.150	72.9 (67.4–78.4) vs. 78.0 (66.8–89.3)	1.6 (0.8–2.9)	0.153	-	-
Tumor grade 3 (vs. grade 1–2)	2.1	0.145	67.6 (61.1–74.1) vs. 85.7 (75.6–95.9)	1.6 (0.8–3.3)	0.210	-	-
Endometrioid histology (vs. other histology)	0.0	0.869	80.3 (71.2–89.4) vs. 69.5 (61.6–77.5)	0.9 (0.4–1.7)	0.656	-	-
NLR ≥ 2.2 (vs. <2.2)	4.8	0.028	66.4 (59.6–73.2) vs. 91.7 (82.5–100.9)	2.2 (1.1–4.7)	0.043	2.3 (1.1–4.6)	0.027
SII ≥ 1100 (vs. <1100)	5.6	0.017	58.5 (44.9–72.1) vs. 82.3 (73.5–91.2)	2.2 (1.1–4.6)	0.047	2.3 (1.2–4.7)	0.017
MLR ≥ 0.18 (vs. <0.18)	7.1	0.008	75.4 (67.2–83.6) vs. 79.4 (74.7–84.1)	5.0 (1.1–20.8)	0.027	5.9 (1.4–24.6)	0.015
Lymphocytes <1.0 (vs. ≥1.0)	10.0	0.002	39.1 (21.6–56.6) vs. 82.1 (74.5–89.7)	3.8 (1.6–9.0)	0.003	4.3 (1.8–10.7)	0.001

OS: overall survival, X^2^: Chi-square test, HR: Hazard Ratio, IC: confidence interval, FIGO: International Federation of Gynecology and Obstetrics, NLR: Neutrophil-to-Lymphocytes ratio, SII: Systemic Immune-Inflammatory Index, MLR: Monocyte-to-Lymphocyte ratio).

**Table 2 jcm-09-01441-t002:** Cancer-specific survival analysis (*n* = 155).

CSS, *n* = 155 Variables	Kaplan-Meier Survival Analysis, CSS, *n* = 155 Patients			UNIVARIATE Cox Regression		MULTIVARIATE Cox Regression	
	X^2^	*p*-Value	CSS (Range)	HR (IC 95%)	*p*-Value	HR (IC 95%)	*p*-Value
Age ≥ 65 years (vs. <65 years)	1.9	0.167	68.5 (60.0–77.0) vs. 89.5 (82.2–96.9)	1.4 (0.8–2.5)	0.206	-	-
FIGO stage III (vs. stage I–II) *	4.5	0.034	78.0 (66.8–89.3) vs. 75.7 (70.4–81.1)	2.0 (1.04–3.9)	0.038	2.2 (1.1–4.3)	0.021
Tumor grade 3 (vs. grade 1–2)	1.6	0.206	69.8 (63.3–76.3) vs. 89.4 (79.7–99.1)	1.1 (0.6–2.1)	0.648	-	-
Endometrioid histology (vs. other histology)	0.2	0.656	86.2 (78.1–94.3) vs. 70.1 (62.9–78.9)	0.8 (0.4–1.4)	0.391	-	-
NLR ≥ 2.2 (vs. <2.2)	4.3	0.038	68.9 (62.0–75.7) vs. 93.0 (84.0–102.1)	1.4 (0.8–2.5)	0.285	-	-
SII ≥1100 (vs. <1100)	4.1	0.042	63.2 (49.5–76.9) vs. 87.7 (80.9–94.5)	1.4 (0.7–2.7)	0.352	-	-
MLR ≥ 0.18 (vs. <0.18)	6.0	0.014	60.5 (47.0–74.1) vs. 79.4 (74.7–88.6)	4.9 (1.2–20.8)	0.027	5.4 (1.3–22.7)	0.020
Lymphocytes <1.0 (vs. ≥1.0)	11.1	0.001	39.2 (21.7–56.6) vs. 87.6 (81.2–94.1)	4.0 (1.7–9.7)	0.002	5.0 (2.0–12.5)	0.001

* FIGO stage III in separate Cox multivariate models: HR = 2.4 (IC 95% 1.2–4.7, *p* = 0.014) for lymphopenia. CSS: cancer-specific survival, X^2^: Chi-square test, HR: Hazard Ratio, IC: confidence interval, FIGO: International Federation of Gynecology and Obstetrics, NLR: Neutrophil-to-Lymphocytes ratio, SII: Systemic Immune-Inflammatory Index, MLR: Monocyte-to-Lymphocyte ratio.

**Table 3 jcm-09-01441-t003:** Progression-specific survival analysis (*n* = 155).

PFS, *n* = 155 Variables	Kaplan-Meier Survival Analysis			UNIVARIATE Cox Regression		MULTIVARIATE Cox Regression	
	X^2^	*p*-Value	PFS (Range)	HR (IC 95%)	*p*-Value	HR (IC 95%)	*p*-Value
Age ≥ 65 years (vs. <65 years)	2.3	0.128	65.7 (59.3–72.1) vs. 61.9 (51.4–72.4)	1.4 (0.8–2.5)	0.206	-	-
FIGO stage III (vs. stage I–II)	**11.4**	**0.001**	**71.3 (65.1–77.5) vs. 54.9 (44.3–65.5)**	**2.3 (1.3–4.1)**	**0.004**	**2.4 (1.4–4.3)**	**0.002**
Tumor grade 3 (vs. grade 1–2)	0.2	0.650	67.2 (59.3–75.0) vs. 66.3 (57.2–75.3)	1.1 (0.6–2.1)	0.648	-	-
Endometrioid histology (vs. other histology)	0.3	0.574	66.5 (59.1–73.8) vs. 66.5 (59.1–73.8)	0.8 (0.4–1.4)	0.391	-	-
NLR ≥ 2.2 (vs. <2.2)	0.7	0.409	62.3 (54.7–69.8) vs. 68.5 (58.9–78.2)	1.4 (0.8–2.5)	0.709	-	-
SII ≥1100 (vs. <1100)	1.7	0.192	67.6 (60.8–74.3) vs. 55.6 (41.6–69.6)	1.4 (0.7–2.7)	0.719	-	-
MLR ≥ 0.18 (vs. <0.18)	3.0	0.083	77.7 (65.8–89.7) vs. 60.7 (54.3–67.1)	2.2 (0.9–5.1)	0.080	-	-
Lymphocytes <1.0 (vs. ≥1.0)	**4.6**	**0.033**	**67.5 (61.3–73.8) vs. 37.2 (18.9–55.5)**	**2.4 (1.1–5.6)**	**0.049**	**2.7 (1.1–6.4)**	**0.024**

PFS: progression-free survival, X^2^: Chi-square test, HR: Hazard Ratio, IC: confidence interval, FIGO: International Federation of Gynecology and Obstetrics, NLR: Neutrophil-to-Lymphocytes ratio, SII: Systemic Immune-Inflammatory Index, MLR: Monocyte-to-Lymphocyte ratio.

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
