# Peer review of "Analysis of Systemic Inflammatory Factors and Survival Outcomes in Endometrial Cancer Patients Staged I-III FIGO and Treated with Postoperative External Radiotherapy"

_jcm, 2020, doi:10.3390/jcm9051441_

Round 1

Reviewer 1 Report

Prognostic impact of systemic inflammatory factors on survival outcomes in endometrial cancer patients staged I-III FIGO and treated with postoperative external radiotherapy.

This paper investigates inflammatory factors (NLR, MLR, SII and lymphopenia) in endometrial cancer prognosis of women from two hospital cohorts and treated with external beam radiotherapy +/- brachytherapy and chemotherapy; essentially any women with stage IB or higher not ‘cured’ with surgery. The paper found that these inflammatory markers were associated with a poorer survival using Kaplan Meier, univariate and multivariate models.

Overall comment:

This study bases its novelty on the ‘homogenously treated cohort’ and states that no inflammatory biomarkers have been proposed to regime the EC risk assessment. However, in their study, the authors find similar results to what is already published (Cummings 2015, Hannuma 2015, Ethier 2017, Acikgov 2017) and do not discuss how their results could be included in risk assessment.

I am surprised that either stage, grade or histology did not reach significance as prognostic factors, even though an attempt to justify is due to the cohort choice. To me this indicates something skewed in the cohort and makes me question the results.

From a quick literature search, there are a few key and recent papers not mentioned which I think needs to be discussed:

  • Mirili, Cem, and Mehmet Bilici. "Inflammatory prognostic markers in endometrial carcinoma: Systemic immune-inflammation index and prognostic nutritional index." Medical Science and Discovery 1 (2020): 351-359.
  • Dong, Yangyang, Yuan Cheng, and Jianliu Wang. "The ratio of neutrophil to lymphocyte is a predictor in endometrial cancer." Open Life Sciences 1 (2019): 110-118.
  • Aoyama, Tadashi, et al. "Pretreatment neutrophil-to-lymphocyte ratio was a predictor of lymph node metastasis in endometrial cancer patients." Oncology 5 (2019): 259-267.

Specific comments:

  1. Please give details of where the patient cohorts were obtained from (two European institutions?).
  2. Were there patients treated with brachytherapy but not adjuvant EBRT and excluded? Should these have been included?
  3. Should patients treated with EBRT/brachy and those with chemotherapy be split into separate groups?
  4. Figure S1: why would patients be diagnosed with EC by biopsy but not have surgical treatment? Please explain abbreviations ie IGR.
  5. BMI should be included in analysis as this is a major factor in endometrial cancer and inflammation.

Author Response

Dear Reviewer,

Thank you very much for all your comments and suggestions. We have made all the suggested changes (please see the details below), together with some improvements in the language usage. We have also emphasised some concepts such as the role of lymphopenia, by adding more information in the Results and the Discussion sections. Attached you will find the new version of our manuscript.

Kind regards,

Katarzyna Holub

Question 1:This study bases its novelty on the ‘homogenously treated cohort’ and states that no inflammatory biomarkers have been proposed to regime the EC risk assessment. However, in their study, the authors find similar results to what is already published (Cummings 2015, Hannuma 2015, Ethier 2017, Acikgov 2017) and do not discuss how their results could be included in risk assessment.

Response 1:We have added more details from these studies in the introdution and thediscussion

Introduction:

Nevertheless, in the case of EC, there is insufficient evidence to support their use in clinical practice, as shown by Ethier et al. in an extensive review of 26 studies with a total of 10,530 patients with different gynecological cancers published in 2017, where EC was depicted only in two studies with less than 1000 patients  [Ethier, Haruma, Cummings].

Discussion:

Acikgoz et al. reported that the preoperative NLR>2.41 was a significant predictor for cervical stromal involvement in endometrioid endometrial adenocarcinoma (p=0.006, OR= 2.03) [Acigoz].In an article based on the retrospective data of 320 patients, Hannuma et al. confirmed that pre-treatment NLR was an independent predictor of poor prognosis in EC (HR 3.3; 95% CI 1.2-9.5; p=0.026) [18]. In a systematic review of eleven studies, elevated NLR was related with advanced stage of disease in EC patients [Hannuma].

The strength of our study in comparison with previously reported publications relies on its bi-institutional cohort of high-risk patients, all of them treated with EBRT. Until now, the only multicenter study with a larger cohort than ours, with 605 patients, was published by Cummings et al., but only 33% were treated with radiotherapy. Cummings et al. also included patients in FIGO stage IV [Cummings], while all our patients were staged as FIGO I-III, but classified within the ESMO-ESGO ESTRO 3-5 groups.

Question 2:I am surprised that either stage, grade or histology did not reach significance as prognostic factors, even though an attempt to justify is due to the cohort choice. To me this indicates something skewed in the cohort and makes me question the results.

From a quick literature search, there are a few key and recent papers not mentioned which I think needs to be discussed:

  • Mirili, Cem, and Mehmet Bilici. "Inflammatory prognostic markers in endometrial carcinoma: Systemic immune-inflammation index and prognostic nutritional index." Medical Science and Discovery 1 (2020): 351-359.
  • Dong, Yangyang, Yuan Cheng, and Jianliu Wang. "The ratio of neutrophil to lymphocyte is a predictor in endometrial cancer." Open Life Sciences 1 (2019): 110-118.
  • Aoyama, Tadashi, et al. "Pretreatment neutrophil-to-lymphocyte ratio was a predictor of lymph node metastasis in endometrial cancer patients." Oncology 5 (2019): 259-267.

Response 2:In the discussion we have emphasised the reason for which stage, grade or histology did not reach significance as prognostic factors. Additionally, we have added all these articles in the discussion:

The classically applied variables in our study such as histology (endometrioid vs. others), tumor grade (grade 3 vs. grade 1-2) and FIGO stage III (vs. stage I-II) did not achieve statistical significance in univariate Cox regression. We assume that adjuvant treatments may have alleviated the effects of such clinical features by decreasing their intrinsic prognostic value. Moreover, all our patients belong to the high-risk cohort (ESMO-ESGO-ESTRO 3-5 group), where these classical parameters should not be analysed separately but as a sum of characteristics. For example, the tumor grade 3 is a stronger prognosticator of more aggressive evolution than FIGO stage or histology [4].

These data are in line with other recently published results in the literature. Dong et al. reviewed clinical records of 510 Chinese EC patients surgically treated between 2010-2016 and found that NLR was an independent prognostic marker for OS (HR 4.7; 95% CI, 1.5-14.1; p =0.006), CSS (HR 3.6; 95% CI, 1.1-11.5; p =0.028) and disease free survival (DFS, HR 2.3; 95% CI, 1.0-5.2; p =0.044) [25]. Based on the data of 101 patients, Mirili et al. confirmed that NLR>3.3 and PLR>177 were associated with shorter PFS and OS, and were the first to prove that SII>1035.9 and prognostic nutritional index (PNI) <38 were also independent prognostic factors for worse survival outcomes in EC. The authors also examined a correlation between inflammatory factors and classically used prognosticators such as lymph node involvement, FIGO stage, lymphovascular invasion, and cervical stromal invasion, which they found that were associated with higher NLR, SII, and lower PNI. Moreover, NLR and PNI were associated with ECOG performance scores 2-3 and myometrial invasion [26]. Similarly, a recent Japanese study on 197 EC patients identified a high NLR and PLR as predictive of lymph node involvement, observed in 25 patients (13%) [27].

Question 3:Please give details of where the patient cohorts were obtained from (two European institutions?).

Response 3: I propose to add this information in the abstract: at Gustave Roussy Institut, France, and Hospital Clínic, Spain, two European institutions between 2008 and 2017

Question 4:Were there patients treated with brachytherapy but not adjuvant EBRT and excluded? Should these have been included?

Response 4: I have added this information in the discussion:

Cummings et al. also included patients in FIGO stage IV [17], while all our patients were staged as FIGO I-III but classified within the ESMO-ESGO ESTRO 3-5 groups. For this reason, patients treated with exclusive HRD-BT did not meet inclusion criteria, as in accordance with the guidelines published by ESMO-ESGO-ESTRO in 2015, only patients belonging to the high-intermediate, high and advanced risk group should be treated with postoperative adjuvant EBRT +/- BT (patients with low and intermediate risk of recurrence were excluded from the study) [4].

Question 5:Should patients treated with EBRT/brachy and those with chemotherapy be split into separate groups?

Response 5: I have added this information in the results:

All patients were treated with EBRT +/- BT, and additionally adjuvant chemotherapy (CT) was given to 84 patients (52.2%), but it had no impact on mOS (p=0.379), with 22.6% of deaths in CT cohort vs. 28.6% in non-CT cohort.

Question 6:Figure S1: why would patients be diagnosed with EC by biopsy but not have surgical treatment? Please explain abbreviations ie IGR.

Response 6: All patients included in our study underwent surgery as a first treatment (patients who  was not surgically treated were excluded, as showed in Figure S1). We have explained the abbreviations used in Figure S1 (EC: Endometrial cancer, EBRT: External Beam Radiotherapy, IGR: Institut Gustave Roussy, FIGO: International Federation of Gynecology and Obstetrics).

Question 7.BMI should be included in analysis as this is a major factor in endometrial cancer and inflammation.

Response 7: We have added this information in the discussion:

Unfortunately, we did not have the records of patients’ weight and height for the majority of Spanish patients, and therefore we were unable to assess the relationship between the Body Mass Index (BMI) and inflammatory factors.

Reviewer 2 Report

Katarzyna Holub and collaborators analyzed inflammatory biomarkers in endometrial cancer patients. The study type should be clearly stated (observational, retrospective ….).

The relative importance of inflammation factors in endometrial cancer outcomes can be different between premenopausal and postmenopausal women. In fact, the role of estrogen in inflammation is important (Endocr Relat Cancer. 2010). Although authors addressed age, the relevance of ≥ 65years old does seem appropriate. In addition, although authors states inflammatory biomarkers as indicators of endometrial cancer prognosis the current study misses important markers like CRP, IL-6 or TNF-α.

The sample size is relatively small to assess inflammation markers. Nevertheless, my major concerns regard the contribution of other factors to the observed results. The authors have not adjusted to important factors that may contribute to inflammatory markers instead of endometrial cancer, particular BMI, which is associated to inflammation. Nevertheless, the study is interesting and may be relevant for Journal of Clinical Medicine readers.

Author Response

Dear Reviewer,

Thank you very much for all your comments and suggestions. We have made all the suggested changes (please see the details below), together with some improvements in the language usage. We have also emphasised some concepts such as the role of lymphopenia, by adding more information in the Results and the Discussion sections. Attached you will find the new version of our manuscript. 

Kind regards,

Katarzyna Holub

Question 1.The study type should be clearly stated (observational, retrospective ….).

Response 1. We have added add this information in the abstract:

Here, we examined the role of inflammatory indicators as a tool to identify EC patients at higher risk of death in retrospective observational study.

Question 2.The relative importance of inflammation factors in endometrial cancer outcomes can be different between premenopausal and postmenopausal women. In fact, the role of estrogen in inflammation is important (Endocr Relat Cancer. 2010). Although authors addressed age, the relevance of ≥ 65years old does seem appropriate.

Response 2. We have added in the results:

Regarding the possible influence of age on inflammatory factors, we did not observe any significant impact of age <50 and age <55 on mOS (p=0.125 and p=0.632), nor any significant correlation with levels of NLR, SII, MLR or lymphopenia. Menopausal status was not available, and could therefore not be tested accurately.

Question 3.In addition, although authors states inflammatory biomarkers as indicators of endometrial cancer prognosis the current study misses important markers like CRP, IL-6 or TNF-α.

Response 3. Unfortunately, we had no basal data of CRP, IL-6 and TNF-α.

Question 4.The sample size is relatively small to assess inflammation markers. Nevertheless, my major concerns regard the contribution of other factors to the observed results. The authors have not adjusted to important factors that may contribute to inflammatory markers instead of endometrial cancer, particular BMI, which is associated to inflammation. Nevertheless, the study is interesting and may be relevant for Journal of Clinical Medicine readers.

Response 4. We have added this information in the discussion:

Unfortunately, we did not have the records of patients’ weight and height for the majority of Spanish patients, and therefore we were unable to assess the relationship between the Body Mass Index (BMI) and inflammatory factors.

Round 2

Reviewer 1 Report

The authors have made the required changes to the manuscript, including more discussion around the meaning of their results and how they align with other key recent papers.